# Development of a Genetically Engineered Mouse Model Recapitulating LKB1 and PTEN Deficiency in Gastric Cancer Pathogenesis

**DOI:** 10.3390/cancers15245893

**Published:** 2023-12-18

**Authors:** Kuan-Te Fang, Hsin Hung, Nga Yin Sadonna Lau, Jou-Hsi Chi, Deng-Chyang Wu, Kuang-Hung Cheng

**Affiliations:** 1Institute of Biomedical Sciences, National Sun Yat-Sen University, Kaohsiung 80424, Taiwan; d092050007@student.nsysu.edu.tw (K.-T.F.); m032050004@student.nsysu.edu.tw (H.H.); d118020009@student.nsysu.edu.tw (N.Y.S.L.); d118020013@student.nsysu.edu.tw (J.-H.C.); 2Center of Excellence for Metabolic Associated Fatty Liver Disease, National Sun Yat-Sen University, Kaohsiung 80424, Taiwan; 3Division of Gastroenterology, Department of Internal Medicine, Kaohsiung Medical University Hospital, Kaohsiung Medical University, Kaohsiung 80708, Taiwan; dechwu@kmu.edu.tw; 4Department of Medical Laboratory Science and Biotechnology, Kaohsiung Medical University, Kaohsiung 80708, Taiwan; 5National Institute of Cancer Research, National Health Research Institutes, Tainan 70456, Taiwan

**Keywords:** gastric cancer, mouse models of cancer, intestinal type of gastric cancer, liver kinase B1, phosphatase and tensin homolog, phosphoinositide 3-kinase signaling pathway, E-cadherin

## Abstract

**Simple Summary:**

Gastric cancer (G.C.) is a lethal human malignancy, boasting the highest mortality rate among all cancers. Recent advances in its molecular pathology have unveiled a comprehensive genetic profile associated with the initiation and progression of G.C. These encompass the activation of KRAS, along with the inactivation or loss of E-cadherin, PTEN, and P53, forming the cornerstone for delving into the genetic and molecular underpinnings of this malignancy. We analyzed the effects of the loss of LKB1, an E-cadherin expression regulator, and PTEN, a negative regulator of the phosphoinositide 3-kinase (PI3K) signaling pathway, on G.C. development in mice. Our results indicated that these two genes act as suppressors in gastric carcinogenesis, and the loss of these genes promotes the development of G.C. in mice.

**Abstract:**

The LKB1 and PTEN genes are critical in gastric cancer (G.C.) development. LKB1, a robust tumor suppressor gene, encodes a serine/threonine kinase that directly triggers the activation of AMPK—an integral cellular metabolic kinase. The role of the LKB1 pathway extends to maintaining the stability of epithelial junctions by regulating E-cadherin expression. Conversely, PTEN, a frequently mutated tumor suppressor gene in various human cancers, emerges as a pivotal negative regulator of the phosphoinositide 3-kinase (PI3K) signaling pathway. This study is set to leverage the H+/K+ ATPase Cre transgene strain to precisely target Cre recombinase expression at parietal cells within the stomach. This strategic maneuver seeks to selectively nullify the functions of both LKB1 and PTEN in a manner specific to the stomach, thereby instigating the development of G.C. in a fashion akin to human gastric adenocarcinoma. Moreover, this study endeavors to dissect the intricate ways in which these alterations contribute to the histopathologic advancement of gastric tumors, their potential for invasiveness and metastasis, their angiogenesis, and the evolving tumor stromal microenvironment. Our results show that conditional deletion of PTEN and LKB1 provides an ideal cancer microenvironment for G.C. tumorigenesis by promoting cancer cell proliferation, angiogenesis, and metastasis.

## 1. Introduction

Gastric cancer (G.C.), currently the third leading cause of cancer-related deaths globally and notably prevalent in Asia [1,2], manifests as a solid tumor with intricate interplays between genetics and environment driving its onset and progression. Early-stage stomach cancer often remains asymptomatic, complicating its early detection. Common indications of patients with stomach cancer encompass diminished appetite, weight loss, abdominal pain, anemia, and vomiting. Early G.C. disregards lymph node status, characterized by invasive adenocarcinoma confined to the mucosa or submucosa and not extending to the muscularis propria. Predominantly afflicting males, with a male-to-female ratio of 1.4–2.4:1, G.C. is typically diagnosed around the median age of 63 (range 21–89) years [2]. In advanced G.C., presenting symptoms and signs may diverge in cases with distant metastasis and abdominal enlargement due to liver metastasis or malignant peritoneal effusion, and occasionally, palpable non-regional lymph node metastasis (Virchow’s lymph node) [3,4]. The majority of G.Cs. are adenocarcinomas, traditionally classified into two subtypes based on Lauren’s system—intestinal and diffuse [5,6]. Diffuse G.C. entails poorly differentiated cancer cells amidst stromal cells, whereas intestinal G.C. is characterized by cancer cells forming gland-like tubular structures with limited stromal involvement [7]. G.C. culminates from a confluence of factors, including environmental influences, unstable mucosal homeostasis, hereditary predisposition, gene mutations, and *H. pylori* infection [8]. Hence, we engineered a mouse model of G.C. to probe the alterations in pathogenic signaling pathways and the microenvironment within G.C. [9]. 

The mouse stomach comprises three distinct regions: the forestomach, corpus, and antrum [10]. The forestomach, primarily for food storage, features squamous cells. The corpus, a glandular–oxyntic area, houses diverse cell types such as mucous pit cells, mucous neck cells, chief cells (secreting pepsinogen), and parietal cells (secreting acid). The antrum contains gastrin and enterochromaffin-like cells (ECL) [11,12]. To safeguard against self-digestion, mucous pit cells (expressing MUC5AC and TFF1) and mucous neck cells (expressing MUC6 and TFF2) produce protective mucins [13,14]. Gastric chief cells, expressing MIST1, release pepsinogen and gastric lipase, activated by hydrochloric acid (HCl) [15,16]. Parietal cells, residing in the gastric glands, secrete HCl and intrinsic factor. H+/K+ ATPase, a parietal cell marker, is regulated by ECL, gastrin G, and somatostatin D cells. Notably, parietal cells are exclusive to the corpus region in mouse stomachs, contrasting human stomachs, where they are distributed throughout. Gastric stem cells are thought to reside in the isthmus region, with mucous neck cells expressing TFF2, Sox2, and LGR5 potentially functioning as stem/progenitor cells [17,18,19,20].

LKB1 (liver kinase B1), also recognized as serine/threonine kinase 11 (STK11), encodes an approximately 50 kDa serine/threonine kinase. Positioned on human chromosome 19p13.3, LKB1 mutations have been linked to hereditary Peutz–Jeghers syndrome (PJS) [21,22] LKB1 has a critical role in activating tuberous sclerosis complex 2 (TSC2), another vital tumor suppressor gene frequently found mutated in a wide array of human cancers [23,24]. This gene regulates cell polarity and is often mutated or deleted in various cancer types [25]. LKB1’s array of substrates encompasses the kinase AMPK and 12 other members of the AMPK-like kinase family, including MARKs (microtubule affinity regulating kinases; 1–4), SIKs (salt-inducible kinases; 1–3), BRSKs (brain-specific kinases or SADs; 1 and 2), and NUAKs (1 and 2). These substrates regulate diverse cellular processes such as autophagy, metabolism, cell growth, and cell polarity. However, the extent to which AMPK functions as an LKB1 target in relation to polarity and tumorigenesis remains uncertain [23,25]. The proteins tuberous sclerosis complex 1 (TSC1) and tuberous sclerosis complex 2 (TSC2) operate in tandem to restrain cell growth by negatively modulating the mammalian target of mTOR [26]. Among the principal growth-regulating pathways orchestrated by LKB1-AMPK is the mTOR pathway. mTOR, a central coordinator of growth factor inputs, governs cell growth across eukaryotes and often loses regulation in human cancers [26]. Within the cascade of mitogenic pathways repressed by LKB1-AMPK signaling, the mTOR complex 1 (mTORC1) pathway is curbed through AMPK-mediated phosphorylation of TSC2 [25]. LKB1-AMPK pathway function also includes the maintenance of epithelial junction stability by regulating E-cadherin expression. Previously, we unveiled LKB1’s influence on the Wnt/β-catenin pathway in lung cancer, along with indications that silencing LKB1 leads to heightened epithelial–mesenchymal transition (EMT) markers [21]. Thus, LKB1 assumes a crucial role within the domain of human cancer research.

PTEN (phosphatase and tensin homolog), located on human chromosome 10q23, emerged as a tumor suppressor gene with a pivotal role in negatively regulating the PI3K/AKT/mTOR signaling pathway. This pathway can be set into several biological effects through various mechanisms, all culminating in amplified pathway activation within cancer cells [26]. PI3K comprises the p110 catalytic subunit and the p85 regulatory subunit. By phosphorylating PIP2 (phosphatidylinositol 3,4-bisphosphate), PI3K generates PIP3 (phosphatidylinositol 3,4,5-trisphosphate). Subsequently, PIP3 triggers the activation of PDK1 (3-phosphoinositide-dependent kinase 1) and AKT, fostering cell proliferation and survival. The phosphorylation of Akt, in turn, kindles the activity of mTOR, a cardinal controller of protein translation [27]. PTEN counteracts this cascade by dephosphorylating PIP3, thereby restraining the activation of Akt triggered by PIP3 [28,29]. Recent studies have unveiled a prevalent occurrence of PTEN loss or reduced expression in G.C.s, attributed to genetic or epigenetic alterations like mutations, loss of heterozygosity (LOH), and promoter hypermethylation [30,31].

In this study, we have established cohorts of mutant mice wherein LKB1 alleles are rendered inactive, alongside conditional PTEN null alleles, driven by the stomach-specific H+/K+ ATPase-Cre transgenic line [32]. Through a series of periodic non-invasive imaging studies and consecutive histological examinations, we aim to delineate the kinetics of G.C. development and trace the histopathological evolution within these mutant mouse populations.

## 2. Materials and Methods

### 2.1. Animal Models

We constructed an H+/K+ ATPase-Cre transgenic strain that can express Cre recombinase specific to the stomach’s parietal cells. The H+/K+ ATPase-Cre transgenic mice were generated by the National Laboratory Animal Center (NLAC, Taipei, Taiwan). The LKB1 (FVB;129S6-Stk11tm1Rdp) mice were obtained from the NCI Mouse Repository. The PTEN (Ptentm1Hwu) mice were purchased from The Jackson Laboratory (Sacramento, CA, USA). We further crossed our H+/K+ ATPase-Cre mice with LKB1 and PTEN conidtional floxped alleles miceto induce gastric adenocarcinoma formation in the mice. The animals were housed at the animal center of National Sun Yat-Sen University (NSYSU) in specific pathogen-free (SPF) conditions following the guidelines of the Association for the Assessment and Accreditation of Laboratory Animal Care (AAALC). All the procedures were approved by the NSYSU Institutional Animal Care and Use Committee (IACUC) under approval number 110-07.

### 2.2. Mouse Tail DNA Isolation

To purify mouse DNA, mice tails were lysed with lysis buffer (NaCl, Tris-HCl pH 8.0, 2% SDS, 2 mM EDTA pH 8.0, and 20 mg/mL proteinase K). The samples were incubated at 57 °C for 1 h on a shaker at 150 rpm. After 1 h, 250 mL of 5 M NaCl in each sample was cooled down on crushed ice for 10 min, then centrifuged at 9000× *g* rpm for 15 min. After centrifugation, the supernatant was transferred to a new Eppendorf tube, 650 mL of isopropanol was added, and the solution was cooled on ice for 10 min. The samples were then centrifuged at 13,000 rpm for 10 min. After that, the supernatant was removed and placed in 75% ethanol, then the samples were centrifuged at 13,000 rpm for 10 min. Subsequently, the supernatant was removed to dry. The DNA was resuspended in 150 mL of ddH_2_O.

### 2.3. PCR Genotyping

To confirm the genotypes of the mice, we used 2 µL of DNA, 0.4 µL of 2.5 mM dNTP, 0.4 µL of Taq DNA polymerase, 2 µL of 10× reaction buffer, and 0.5 µL of primer. The reaction protocol was as follows: 94 °C for 3 min, followed by 35 cycles of 94 °C, 65 °C and 72 °C for 30 s at each temperature, then 72 °C for 3 min and hold at 4 °C. The primers used wer as follows: LKB1 (PCRS5: 5′-TCTAACAATGCGCTCATCGTCATCCTCGGC-3′; LKB36: 5′-GGGCTTCCACCT GGTGCCAGCCTGT-3′; LKB39: 5′-GAGATGGGTACCAGGAGTTGGGGCT-3′), PTEN (oIMR9554: 5′-CAAGCACTCTGCGAACTGAG-3′; oIMR9555: 5′-AAGTTTTTG AAGGCAAGATGC-3′), H+/K+ ATPase-Cre (oIMR1084; 5′-GCGGTCTGGCAGTAA AAACTATC-3′; oIMR1085; 5′-GTGAAACAGCATTGCTGTCACTT-3′; oIMR7338: 5′-CTAGGCCACAGAATTGAAAGATCT-3′; oIMR7339: 5′-CTAGGCCACAGAATT GAAAGATCT-3′) [32].

### 2.4. Magnetic Resonance Imaging

The mice underwent anesthesia using 1–2% isoflurane/air, and their body temperature was maintained using air conditioning within the 3T MRI scanner (GE, HDXt Sigma; GE, Milwaukee, WI, USA). The imaging employed a high-resolution animal coil (3.0 cm diameter), with mice placed supine in the coil and securely taped on the bed to minimize respiratory motion. T2-weighted images were obtained using a fast spin echo multi-slice sequence for both coronal (TR/TE 2000/63.23 ms) and axial (TR/TE 5083/46.7 ms) sections. The parameters included 16 echo trains, 4 averages, 2 dummy scans, a field of view of 8 × 4.8 cm^3^ for coronal sections and 6 × 6 cm^2^ for axial sections, a matrix size of 256 × 192, a slice thickness of 2 mm, and 20 contiguous slices. The scans were conducted at 10 min intervals until the 90 min mark. A glass cylinder filled with pure water served as a standard reference for each mouse.

### 2.5. Histology

The mouse stomach tissues were opened and washed with 1× PBS. The tissues were immersed and fixed in 10% formalin and embedded in paraffin. The tissues were sectioned at 5 µL. The sections were deparaffinized and rehydrated in xylene and a graded series of ethanol solutions. The sections were stained with hematoxylin and eosin (H&E) [33,34].

### 2.6. Immunohistochemistry (IHC) and Immunofluorescence (IF)

The stomach sections were deparaffinized and rehydrated in xylene and a graded series of ethanol solutions. The sections were subjected to antigen retrieval (Vector Laboratories) by boiling for 20 min in a microwave oven. The endogenous peroxidase activity was quenched in 3% H_2_O_2_ for 10 min. The sections were blocked for nonspecific binding using goat serum or mouse Ig blocking reagent (Vector Laboratories, Newark, CA, USA). After washing with 1× PBS, the sections were incubated overnight at 4 °C with the primary antibodies. The primary antibodies were probed using a biotinylated secondary antibody and detected using an avidin/biotin detection system (Vector Laboratories). The sections were detected using DAB solution (Vector Laboratories) and counterstained with hematoxylin. The primary antibodies and dilutions were as follows: Cre (Rabbit; Novus Biologicals, Centennial, CO, USA; 1:200), Ki67 (Rabbit; Abcam, Waltham, MA, USA; 1:100), TGFβ1 (Rabbit; Abcam; 1:100), IL-6 (Rabbit; Santa Cruz, Santa Cruz, CA, USA; 1:50), H+/K+ ATPase (Mouse; Santa Cruz; 1:100), LKB1 (Mouse; Santa Cruz; 1:50), CD133 (Rabbit; Abnova, Taipei, Taiwan; 1:100), α-SMA (Rabbit; Abcam; 1:100), vimentin (Mouse; Santa Cruz; 1:50), PTEN (Rabbit; Genetax; 1:100), EpCAM (Mouse; Santa Cruz; 1:50), CD31 (Rat; Optionen, Cambridge, UK; 1:100), CK7 (Mouse; Santa Cruz; 1:50), CK20 (Mouse; Santa Cruz (E9); 1:50), MMP9 (Mouse; Santa Cruz; 1:50), E-cadherin (Rat; Invitrogen, Carlsbad, CA, USA; 1:100), Sox2 (Rabbit, Abcam (ab97959); 1:200), TFF2 Rabbit, Abcam; 1:200), GSII (Rabbit, Vector; 1:200), LGR5 (Rabbit; Invitrogen; 1:200), C-Kit (Mouse; Cell Signal, Danvers, MA, USA; 1:400), CD44 (Mouse; Cell Signal; 1:400). Alcian blue staining and immunofluorescence were performed as described previously [34].

### 2.7. Fecal Occult Blood Test

Fecal occult blood testing was carried out using the Hemoccult Single Slides diagnostic kit from Beckman Coulter. Following the manufacturer’s instructions, a thin smear of fresh feces was applied to the two test boxes on the card using an applicator stick. The sealed test card was then allowed to incubate for 5 min at room temperature. After the incubation period, a developer was applied to each test spot, including the positive and negative controls on the card.

### 2.8. Blood Chemistry

At the indicated weeks of age, blood samples (0.5–1.0 mL/mouse) were collected for the determination of blood urea nitrogen (BUN) concentrations. Alanine aminotransferase (ALT) and aspartate aminotransferase (AST) activities and complete blood counts (CBC) were measured using a simple measurement device (DRICHEM Fujifilm Medical Co., Ltd., Tokyo, Japan).

### 2.9. Statistical Analysis 

The results are expressed as the means ± S.E. from at least three independent experiments. The statistical significance of the differences between the groups was determined using Student’s *t*-tests (*p* < 0.05).

## 3. Results

### 3.1. Conditional Knockout of LKB1 and PTEN Leads to the Development of Gastric Cancer in Mice

In this study, we generated cohorts of mutant mice to explore the interplay of tumor suppressor genes in gastric tumorigenesis. Through conditional deletion of LKB1 alleles and the implementation of conditional PTEN null alleles using the stomach-specific H+/K+ ATPase-Cre transgenic line, we aimed to investigatethe combined impact of LKB1 and PTEN loss on the development of gastric carcinoma [35,36]. Specifically, we directed the expression of Cre recombinase to the parietal cells of the gastric epithelium, effectively targeting LKB1 and PTEN gene expression within the stomach. We employed conditional LKB1^L/L^ mice, where loxP sites surround exons 3–6 of the LKB1 gene, sourced from MMHCC. Additionally, PTEN^L/L^ mice, featuring loxP sites flanking exon 5, were purchased from the Jackson Laboratory and incorporated into this study (Figure 1A). Next, we bred the LKB1^L/L^ PTEN^L/L^ mice with a mouse strain that specifically expresses Cre recombinase in the parietal cells of the stomach, yielding H+/K+ ATPase-Cre; LKB1^L/L^; PTEN^L/L^ compound mice (Figure 1A). The H+/K+ ATPase (Atp4b) promoter directing the expression of Cre recombinase in the gastric parietal cells of the transgenic mice was also confirmed using immunohistochemistry (IHC) staining with antibodies against Cre recombinase (Figure 1B). The genotyping confirmations were established through specific PCR analysis of mouse tail DNA (Figure 1C). The details of these genotyping analyses are provided in the Materials and Methods section. In our cohort of aging compound mice, we observed that the H+/K+ ATPase-Cre; LKB1^L/L^; PTEN^L/L^ mice displayed a gradual decline in body weight, with a limited survival period of 30 weeks compared to H+/K+ ATPase-Cre; LKB1^L/L^ or PTEN^L/L^ alone and wild-type mice (Figure 1D,E).

Notably, at approximately 20 weeks, the H+/K+ ATPase-Cre; LKB1^L/L^; PTEN^L/L^ mice started to exhibit gastric hyperplasia, with a strikingly reduced survival time of less than 40 weeks, whereas the H+/K+ ATPase-Cre; LKB1^L/L^ and H+/K+ ATPase-Cre (wt) mice survived for more than 65 weeks without encountering any gastric lesions (Figure 1D and Figure 2A). In the realm of clinical diagnosis, magnetic resonance imaging (MRI) analysis has proven to be a valuable non-invasive approach for gross image analysis [37]. MRI imaging of our H+/K+ ATPase-Cre; LKB1^L/L^; PTEN^L/L^ mice unveiled a substantial enlargement of the stomach, accompanied by thickening of the gastric wall (Figure 2(Aii)). Following these investigations, autopsies provided concrete confirmation of several significant findings within the mice under scrutiny. First and foremost, the autopsies definitively ascertained the presence of enlarged stomachs among the mice. This finding held profound implications, suggesting possible disruptions in normal gastric function. A particularly striking observation during the autopsies was the emergence of gastric hyperplastic polyps in a significant proportion of the examined H+/K+ ATPase-Cre; LKB1^L/L^; PTEN^L/L^ mice (Figure 3(Aiii)). In addition to these findings, approximately 50% of the mice exhibited signs of gastric bleeding during the examinations. Intriguingly, the tails of the H+/K+ ATPase-Cre; LKB1^L/L^; PTEN^L/L^ mice displayed a light pink color as they aged (Figure 2(Avi)). Blood samples collected via cardiac puncture revealed a significant reduction in red blood cell count, coupled with chylomicronaemia from the blood hematology analysis, confirming the presence of anemia and hyperlilipdima (Figure 2B) [38]. Meanwhile, serum samples collected from the H+/K+ ATPase-cre; LKB1^L/L^; PTEN^L/L^ mice confirmed chylaemia compared to the wild-type samples. (Figure 2(Aiv)). Haematology assessments indicated a dramatic 300-fold decrease in RBC count, signifying anaemia. A 4.8-fold increase in the WBC count indicated that the mice had undergone an inflammatory state, while the platelet count suggested the potential for hemorrhaging (Figure 2B). Furthermore, fecal occult blood tests from the H+/K+ ATPase-Cre; LKB1^L/L^; PTEN^L/L^ mice exhibited an 80% positive incidence rate at 25 weeks (Figure 2(Av)). Overall, among the H+/K+ ATPase-Cre; LKB1^L/L^; PTEN^L/L^ mice, 75% of the compound mice displayed anemia, 47% of the compound mice developed hematoma, and 68% of the compound mice had tumors that had invaded the duodenum.

To conduct a thorough examination of the histopathological analysis, we further performed a comparative histological analysis using H&E staining on wild-type, H+/K+ ATPase-Cre; PTEN^L/L^, H+/K+ ATPase-Cre; LKB1^L/L^; PTEN^L/+^, and H+/K+ ATPase-Cre; LKB1^L/L^; PTEN^L/L^ mice at approximately 15, 20, and 25 weeks old (Figure 3). Our study revealed distinct patterns in the expression and organization of parietal cells within the corpus and antrum regions of the wild-type mice. Conversely, the mice with the H+/K+ ATPase-Cre; PTEN^L/L^ genetic modification displayed slightly disrupted tissue organization, sporadically accompanied by the presence of transformed cells, while still retaining identifiable parietal cells. Consistently, the mice carrying the H+/K+ ATPase-Cre; LKB1^L/L^; PTEN^L/L^ genetic profile exhibited gastric wall thickening, indicative of hyperplasia. The histology of the H+/K+ ATPase-Cre; LKB1^L/L^; PTEN^L/L^ mice displayed substantial enlargement and thickening of the stomach wall, alongside the development of hyperplastic polyps and submucosal hemorrhage throughout the corpus, including invasion of the duodenum (Figure 3). Additionally, the gastric neoplastic lesions exhibited disorganized structures, deviating from their original architecture, marked by the infiltration of inflammatory immune cells and heightened angiogenesis. Remarkably, the population of normal parietal cells was significantly reduced within the gastric neoplasms of H+/K+ ATPase-Cre; LKB1^L/L^; PTEN^L/L^ compound mice (Figure 3).

### 3.2. H+/K+ ATPase-Cre LKB1^L/L^ PTEN^L/L^ Mice Develop Intestinal-Type Gastric Adenocarcinomas

In order to verify the absence of LKB1 and PTEN protein expression in the stomach, we conducted immunofluorescence (IF) analyses by comparing wild-type mice with a genetically modified strain denoted as H+/K+ ATPase-Cre; LKB1^L/L^; PTEN^L/L^ mice (Figure 4A,B). Our findings demonstrated a significant depletion of both LKB1 and PTEN proteins within the stomach tissues of the H+/K+ ATPase-Cre; LKB1^L/L^; PTEN^L/L^ mice. This strongly indicates the functional inactivation of the LKB1 and PTEN genes in the stomach tissues of these mice. Subsequently, we delved into cell-specific markers for a comparative assessment between the wild-type and H+/K+ ATPase-Cre; LKB1^L/L^; PTEN^L/L^ mice (as shown in Figure 4C). Our data exhibited a decrease in the number of parietal cells (H+/K+ ATPase positive) in the corpus of the H+/K+ ATPase-Cre; LKB1^L/L^; PTEN^L/L^ mice, accompanied by elevated expression of Griffonia simplicifolia II (GSII) in the lower portion of the corpus gland (Figure 4C). Previous research has identified GSII as a marker for mucus neck cells, potentially possessing stem/progenitor cell characteristics [39]. Intriguingly, GSII also serves as an indicator of spasmolytic polypeptide-expressing metaplasia (SPEM), a condition known to be associated with G.C. progression [40]. The link between SPEM and cancer suggests that the observed decrease in parietal cells and the concurrent increase in GSII expression might signify a shift in the stomach microenvironment that could facilitate tumorigenesis [41].

In recent years, gastric adenocarcinoma classification, as defined by the WHO, has primarily adopted the Lauren classification system. This system distinguishes between intestinal type and diffuse type carcinomas, serving as the predominant classification scheme globally [42]. Cytokeratins (CKs), which are intermediate-sized cytoskeleton filaments that are primarily present in epithelial cells, have emerged as crucial markers in this context. The presence of cytokeratin is a robust indicator of both normal epithelial cells and their malignant counterparts, making it a hallmark of epithelial malignancies [43,44]. A commonly used diagnostic approach for classifying subtypes of G.C.s involves examining the expression patterns of cytokeratin 7 (CK7) and cytokeratin 20 (CK20) [45]. These cytokeratin markers play a pivotal role in helping to distinguish and categorize different subtypes of gastric adenocarcinomas, aiding clinicians and researchers in their efforts to characterize and understand this complex disease. In the context of normal gastric epithelium, CK7 is not typically present, whereas CK20 can be observed within the superficial foveolar epithelium (as seen in Figure 5). Our observations in the H+/K+ ATPase-Cre; LKB1^L/L^; PTEN^L/L^ mice indicated an increased expression of CK7 within the gastric gland and neoplastic regions. Additionally, there was an extension of CK20 expression to the lower lesions. Moreover, the Alcian blue (pH 2.5) mucin stain specifically detects acidic mucin, presenting a positive result for intestinal-type G.C. [46]. Notably, the H+/K+ ATPase-Cre; LKB1^L/L^; PTEN^L/L^ mice exhibited strong Alcian blue staining in the lower region and isthmus of the gland lesions, setting them apart from wild-type mice stomach (Figure 5). Remarkably, Alcian blue positivity was also observed at the margins of the invasive lesions. Integrating the histological data and Alcian blue staining, we suggest that the gastric neoplasms originating from the H+/K+ ATPase-Cre; LKB1^L/L^; PTEN^L/L^ mice more closely resemble the human intestinal type of gastric adenocarcinoma.

### 3.3. PTEN and LKB1 Loss Promoted Cell Proliferation and Increased Inflammatory Cytokine Expression in the Stomach

Ki67, a widely recognized indicator of cell proliferation, demonstrated a gradual expansion of positive signals in the H+/K+ ATPase-Cre, LKB1^L/L^; PTEN^L/L^ mice with increasing ages (as depicted in Figure 6A). Interleukin-6 (IL-6) and transforming growth factor-beta 1 (TGF-β) are well-recognized proinflammatory cytokines with significant implications for various biological processes [47]. Meanwhile, in the presence of IL-6, TGF-β1 has been implied to prompt the differentiation of T helper cells, contributing to escalated inflammation and exacerbating autoimmune conditions [48]. This process has far-reaching consequences, including heightened inflammation, enhanced carcinogenesis, and potentially exacerbated autoimmune conditions. Our findings were striking, as we observed pronounced expression of TGF-β1 and IL-6 in both the upper and lower regions of the corpus glands, extending even to the invasion zone, as depicted in Figure 6B. These findings strongly suggest that the loss of two crucial tumor suppressor genes, LKB1 and PTEN, not only fosters unbridled cell proliferation but also triggers a significant and noteworthy inflammatory response within the murine stomach.

### 3.4. LKB1 and PTEN Loss Upregulated the Expression of EMT Markers in Gastric Lesions

Meanwhile, the presence of elevated concentrations of proinflammatory cytokines like IL-6 and TGF-β1 has been linked to an enhancement in the epithelial–mesenchymal transition (EMT) program. Our histological examination of the H+/K+ ATPase-Cre, LKB1^L/L^; PTEN^L/L^ mice unveiled significant invasion and metastasis, notably extending into the duodenum, as illustrated in Figure 3. Concurrently, our molecular investigation into EMT markers yielded compelling results, with substantial upregulation in the expression of alpha-smooth muscle actin (α-SMA), vimentin, and MMP9, alongside a decrease in E-cadherin expression, as depicted in Figure 7. The LKB1 pathway is known for its pivotal role in maintaining cellular polarity, and its intricate interplay with E-cadherin and cellular polarity has been suggested to be a crucial factor in tumorigenesis. A loss of LKB1 and E-cadherin may indirectly contribute to gastric tumorigenesis through this crosstalk [49]. Furthermore, the presence of EpCAM can disrupt E-cadherin-mediated cell–cell adhesion by interfering with the connection between alpha-catenin and F-actin, thereby loosening normal inter-cellular adhesion [50]. In addition, matrix metalloproteinases (MMPs) play pivotal roles in tumor invasion and dissemination. Recent investigations have emphasized the importance of MMP-2 and MMP-9 in breaking down the basement membrane’s type IV collagen, a crucial mechanism for metastasis in G.C. [51,52]. In the context of the LKB1 and PTEN loss G.C. model, the expression of MMP9 was notably increased, especially in the gastric tumor and invasive regions of the H+/K+ ATPase-Cre, LKB1^L/L^; PTEN^L/L^ mice. Notably, our observations also indicated an elevated presence of the angiogenesis marker CD31 within the neoplastic lesions of the H+/K+ ATPase-Cre, LKB1^L/L^; PTEN^L/L^ mice, as shown in Figure 7. This implies that there was an increased angiogenic phenotype during gastric carcinogenesis in these mutant mice. 

### 3.5. H+/K+ ATPase-Cre; LKB1^L/L^; PTEN^L/L^ Downregulated SOX2 and Upregulated the Expression of Stem Cell Markers in Mice Gastric Tissues

Normally, SOX2 exhibits expression within certain differentiated gastric epithelia, which may play a crucial role in sustaining the cell population of the gastric epithelium. Notably, the inactivation or loss of SOX2 expression has been linked to the development of highly invasive and malignant tumors [53]. Another important factor, TFF2, is responsible for maintaining markers of gastric epithelium cells, primarily found in the mucous neck cells of isthmus corpus glands. This molecule likely contributes to safeguarding and repairing the gastrointestinal mucosal lining [54]. Conversely, the progression from a normal gastric gland to intestinal metaplasia (IM) mucosa involves the diminishing or loss of expression of SOX2 and TFF2 [55]. In the stomach wall of H+/K+ ATPase-Cre; LKB1^L/L^; PTEN^L/L^ mice, our observations indicated a significant reduction in SOX2 (Figure 8A) and TFF2 (Figure 8B) within the corpus gland compared to the wild-type mice. These findings suggested that the H+/K+ ATPase-Cre; LKB1^L/L^; PTEN^L/L^ mice experienced a disruption in the normal homeostasis of gastric epithelial cells, initiating the carcinogenic process within the stomach and decreasing the protective qualities of the gastric mucosa. A recent study demonstrated that LGR5+ cells are located at the basal regions of the gastric gland. Meanwhile, LGR5 has been reported to be a biomarker for both adult stem cells and cancer stem cells (CSCs) in the gastrointestinal tract. The Wnt/β-catenin signaling pathway has maintained self-renewal in various types of somatic stem cells, like LGR5+ stem cells in the stomach [56]. LGR5 may function to amplify Wnt signaling in the regulation of gastric progenitor activities [57,58]. Increased LGR5 protein expression may lead to enhancements in the Wnt/β-catenin signaling pathway and promote tumorigenesis and tumor progression. In this study, we also observed increased expression of LGR5 in the gastric tumors derived from the H+/K+ ATPase-Cre; LKB1^L/L^; PTEN^L/L^ mice (Figure 8A) [32,59].

Furthermore, our IHC data also revealed an increased expression of cancer stemness markers CD44, CD133, and c-kit within the corpus gland of the H+/K+ ATPase-Cre; LKB1^L/L^; PTEN^L/L^ mice (Figure 8B). Previous studies have established the significance of cancer stem cells in promoting cancer progression. Common markers like CD44, CD133, and c-Kit, in conjunction with the WNT and Notch signaling pathways, are associated with cancer stemness activities [15,60]. In cases of G.C., clinical reports frequently highlight upregulation of the cancer stemness markers CD44 and CD133, along with an increase in DCLK-1 expression within gastric carcinomas [58,61].

## 4. Discussion

G.C. ranks as the fifth most prevalent cancer globally, with a particularly formidable presence in East Asia marked by persistently high incidence and mortality rates. This malignancy’s diagnostic challenges often relegate detection to advanced stages, resulting in a dismal 5-year survival rate of nearly 60%. The study delves into the pivotal role of LKB1, known for the germline mutations responsible for autosomal dominant Peutz–Jeghers syndrome (PJS), characterized by the development of gastric hamartomas and polyps [22,62]. Moreover, diminished LKB1 expression in clinical settings correlates with poorer cancer prognosis, emphasizing its significance in tumor progression. To unravel the intricacies of gastric carcinogenesis, a mouse model was meticulously established. Notably, a specific Cre recombinase was engineered to localize within the parietal cells’ H+/K+ ATPase in the stomach, a critical component responsible for gastric acid secretion belonging to the P2-type ATPase family [32,35]. This innovative approach resulted in the creation of an H+/K+ ATPase Cre transgenic strain, employing the promoter of the mouse subunit of the H (+)-, K(+)-ATPase gene (Atp4b) to govern tissue-specific excision of the LKB1 and PTEN alleles in mice [32,36]. The outcome of this genetic manipulation was the development of gastric hyperplastic polyps, typically emerging after an average of 20 weeks, eventually progressing to adenocarcinomas, leading to the demise of the mice approximately 40 weeks post-creation [32]. Intriguingly, these compound mice exhibited anemia characteristics, such as pale ears and tails, with an incidence of 75% noted after reaching 25 weeks of age. Notably, prior studies have posited that the loss of the H+/K+ ATPase alpha subunit may predispose individuals to iron-deficiency anemia, mirroring the observed phenomenon in these mice [63]. Of note, our results revealed that the H+/K+ ATPase-Cre; LKB1^L/L^; PTEN^L/+^ mice also developed anemia features at approximately 50 weeks, but this was not found in the H+/K+ ATPase-Cre; LKB1^L/L^ mice until over 70 weeks. Furthermore, these findings parallel clinical investigations, where anemia has been documented in human G.C. patients, underscoring the relevance of the mouse model to the clinical context and the intricate interplay of the factors contributing to G.C.’s multifaceted nature.

The LKB1 gene is vital for embryonic development, with its deficiency in mice causing vascular abnormalities and mid-gestation lethality [64]. Lkb1+/− mice predominantly develop polyps in the glandular stomach and small intestine, underscoring its role as a gastrointestinal tumor suppressor [65]. These mice eventually face complications like intestinal obstruction or bleeding from polyps, occurring before carcinomatous changes or metastasis. 

LKB1 functions as a tumor suppressor in lung, breast, and pancreatic cancers [66]. Recent G.C. research highlights significantly reduced LKB1 expression in cancer tissues compared to adjacent normal mucosae [67]. Elevated LKB1 expression enhances chemosensitivity to 5-FU treatment in G.C. cells and reduces CD44 levels, a cell-surface glycoprotein involved in cell interactions, adhesion, and migration [68]. Similarly, PTEN mutations are frequently found in various cancers [69]. Deleting the Pten gene in mice leads to embryonic lethality between days 6.5 and 9.5 of gestation. Clinical studies have revealed PTEN mutations in 10.6% of advanced G.C. cases, with 55.9% resulting in inactivation, particularly with nonsense mutations [70]. PTEN protein expression gradually diminishes throughout gastric carcinoma progression, correlating strongly with adverse outcomes, including gastric tumor lymph node metastasis, advanced stages, and poor prognoses [70]. Here, we demonstrated that H+/K+ ATPase-Cre; LKB1^L/L^; PTEN^L/L^ mice exhibit hyperplastic polyps and hematoma on the corpus and antrum and metastasis to the duodenum after 25 weeks of age. Interestingly, parietal cells only appeared on the corpus region, a defect in the mice’s antra. However, the neoplasms also presented on the antrum region, suggesting that stomach homeostasis was defective. 

Gastric homeostasis is associated with gastric stem cells. Some studies have indicated that gastric cells are differentiated from one progenitor/stem cell localized on the isthmus region of the corpus gland and the basal region of the antrum gland [39]. LGR5+ gastric stem cells can differentiate into different cell types and are known for their bidirectional migration within the stomach. These stem cells are found in both the intestine and the pyloric stomach, playing a crucial role in maintaining and regenerating epithelial tissues in these areas. Notably, overexpression of LGR5 has been observed in clinical gastric carcinoma, indicating its potential involvement in G.C. development. Meanwhile, some studies have identified other gastric progenitors including Villin and Mist1+ at the base of the gastric glands. The Villin protein gives rise to all gastric cell types, and Mist1+ has been identified as a chief cell marker. In addition, Troy+ chief cells also act as reserve stem cells [71]. TFF2 has also been identified as a progenitor marker that is generally expressed in the corpus region, giving rise to only mucous neck cells, chief cells, and parietal cells. TFF2 can also be a G.C. marker, and loss of TFF2 expression might be an early event of the multi-step process of gastric carcinogenesis. It may also play a limited role in the mucosal protection of the normal gastric physiology. However, despite 30% of G.C. cases being positive for TFF2, a significantly increased expression of TFF2 was noted mainly in large tumors of the diffuse type and advanced G.C. with metastasis [54]. 

In clinical settings, G.C. frequently metastasizes to the lymph nodes (10–20%), the liver (48%), and the peritoneum (32%) [72,73]. In our H+/K+ ATPase-Cre; LKB1^L/L^; PTEN^L/L^ mouse model, we observed distinct pathological manifestations, primarily liver ischemia and necrosis, along with pancreas and spleen inflammation. We also observed that the H+/K+ ATPase-Cre; LKB1^L/L^; PTEN^L/L^ mice exhibited elevated levels of epithelial–mesenchymal transition (EMT) markers through immunohistochemical analyses (Figure 7). These demonstrated a remarkable invasive capacity, as evidenced by their ability to infiltrate the gastric wall and metastasize to the duodenum (Figure 3) [32]. This highly invasive phenotype may be attributed to the dysregulation of EMT in G.C. EMT encompasses a complex cellular process in which gastric epithelial cells acquire mesenchymal characteristics and lose epithelial traits, including the crucial E-cadherin, as well as the upregulation of matrix metalloproteinases (MMPs), vimentin, and α-smooth muscle actin (α-SMA), all contributing to enhanced cancer invasion and metastasis. In addition, our previous study demonstrated that inactivation of LKB1 modulates the Wnt/β-catenin pathway in lung cancer [22]. The Wnt/β-catenin signaling pathway has maintained self-renewal in various types of somatic stem cells, like LGR5+ stem cells in the stomach. LGR5 functions to amplify Wnt signalling in these stem cells [72]. Abnormal activation of the Wnt/β-catenin signaling pathway strongly correlates with tumor progression and metastasis through the maintenance of cancer stemness activities [60,73,74]. These findings illuminate a distinctive metastatic pattern and underscore the significance of EMT and stem cell-like characteristics in facilitating G.C. metastasis. In addition, recent research has emphasized the significant role of inflammation in the progression of tumors. Within the tumor microenvironment, inflammation contributes to several crucial aspects of cancer development, including the proliferation and survival of malignant cells, the stimulation of angiogenesis and metastasis, and the undermining of adaptive immune responses. Inflammation also intricately interacts with numerous signaling pathways, including TGFβ and Wnt [75]. In our study, we have observed pronounced inflammation in mice with compound loss of LKB1 and PTEN, marked by increased levels of TGFβ and the inflammation marker IL-6 within the tumor microenvironment [32]. These findings illuminate the complex interplay between inflammation and cancer progression, highlighting inflammation as a potential driver of immune evasion and a critical target for therapeutic strategies in the context of G.C.

## 5. Conclusions

The conditional abolition of the LKB1 and PTEN genes leads to a cascade of events that disrupt stomach homeostasis. This disruption includes the induction of tumorigenesis, the promotion of epithelial–mesenchymal transitions (EMTs), the upregulation of cancer stemness, and the creation of a pro-inflammatory microenvironment conducive to carcinogenesis. Ultimately, the loss of PTEN and LKB1 in mouse stomach tissues drives carcinogenesis, facilitates tumor invasion, and promotes angiogenesis. These collective processes culminate in the development of gastric intestinal type adenocarcinoma in mice. These findings provide valuable insights into the molecular mechanisms underlying G.C. development and suggest potential targets for therapeutic interventions in the future.

## Figures and Tables

**Figure 1 cancers-15-05893-f001:**
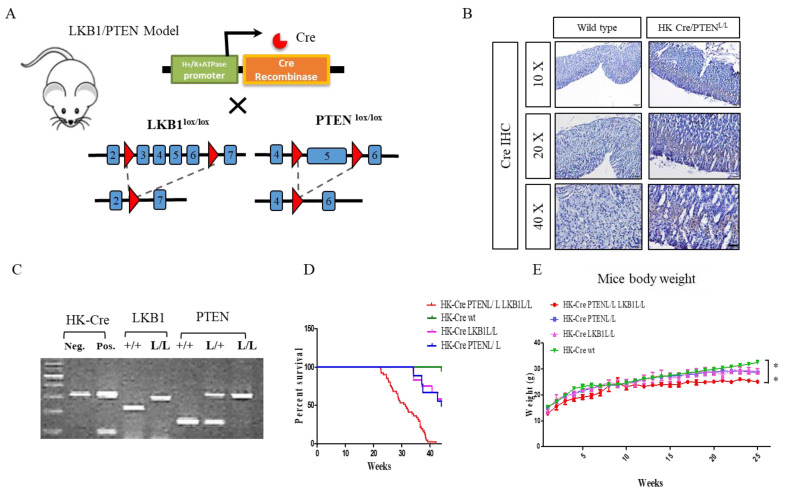
H+/K+ ATPase Cre-mediated conditional deletions of LKB1 and PTEN in the mouse stomach. (**A**) Genetically modified mice: LKB1^L/L^ PTEN^L/L^ mated with H^+^/K^+^ ATPase-Cre LKB1^L/L^ PTEN^L/L^ mice to obtain H^+^/K^+^ ATPase-Cre LKB1^L/L^ PTEN^L/L^ littermates. (**B**) Immunohistochemical analysis for Cre recombinase staining in the stomach tissues of H^+^/K^+^ ATPase-Cre PTEN^L/L^ and wild-type mice. Scale bar: 100 µm. (**C**) Specific genotyping PCR analyses used to detect LKB1, PTEN, and loxP alleles. HK-Cre: Cre(+) = 100 bp, LKB1: wild type = 220 bp/LoxP = 300 bp, PTEN: wild type = 156 bp/heterozygote = 156 bp and 328 bp/LoxP allele = 328 bp. (**D**) Kaplan–Meyer curve showing significantly reduced survival time for H^+^/K^+^ATPase-Cre; LKB1^L/L^; PTEN^L/L^ mice compared with H^+^/K^+^ATPase-Cre; PTEN^L/L^, H^+^/K^+^ ATPase-Cre; LKB1^L/L^ and H^+^/K^+^ATPase-Cre wild-type (wt) mice. (**E**) Measurements of mouse body weight at the indicated time points from the H^+^/K^+^ ATPase-Cre; LKB1^L/L^; PTEN^L/L^, H^+^/K^+^ ATPase-Cre; LKB1^L/L^; PTEN^L/+^, and H^+^/K^+^ ATPase-Cre; LKB1^L/L^ and H^+^/K^+^ ATPase-Cre wild-type mice. ** *p* < 0.01.

**Figure 2 cancers-15-05893-f002:**
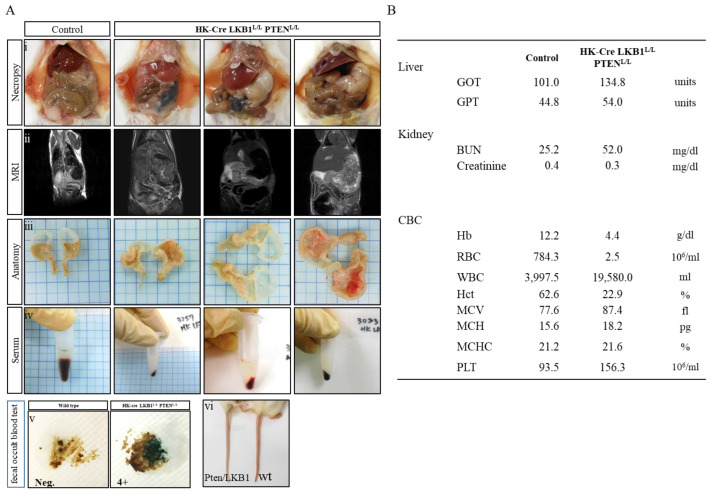
LKB1 and PTEN collaborate in the suppression of G.C. in mice. (**A**) (**i**), Representative in situ abdominal organs of the necropsy analysis of H+/K+ ATPase-Cre; LKB1^L/L^; PTEN^L/L^ and wild-type (control) mice. Severely enlarged stomachs were found in the H+/K+ ATPase-Cre; LKB1^L/L^; PTEN^L/L^ mice. (**ii**), MRI images revealing enlarged stomachs and gastric wall thicknesses in H+/K+ ATPase-Cre; LKB1^L/L^; PTEN^L/L^ mice. (**iii**), Representative gastric stomach of H+/K+ ATPase-Cre; LKB1^L/L^; PTEN^L/L^ animals showing gastric hyperplastic polyps and hemorrhage. (**iv**), Milky-white appearance of plasma samples in H+/K+ ATPase-Cre; LKB1^L/L^; PTEN^L/L^ mice. (**v**), Stool occult blood test identified as 4+ positive in the H+/K+ ATPase-Cre; LKB1^L/L^; PTEN^L/L^ mice. (**vi**), H+/K+ ATPase-Cre; LKB1^L/L^; PTEN^L/L^ mice are anemic, as evidenced by the pale skin and tail color compared to wild-type mice. (**B**) Hematology analysis revealed anemia and inflammation in the H+/K+ ATPase-Cre; LKB1^L/L^; PTEN^L/L^ mice.

**Figure 3 cancers-15-05893-f003:**
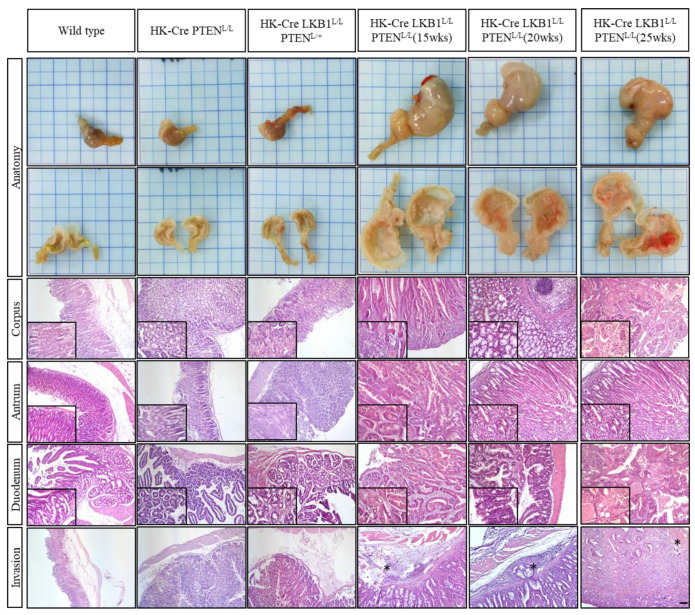
Comparison of the gross morphology and histology of stomach tissues among wild-type, H+/K+ ATPase-Cre; PTEN^L/L^, H+/K+ ATPase-Cre; LKB1^L/L^; PTEN^L/+^, and H+/K+ ATPase-Cre; LKB1^L/L^; PTEN^L/L^ mice. Gross appearances and necropsy images of the stomach isolated from H+/K+ ATPase-Cre; LKB1^L/L^; PTEN^L/L^ mice compared with gastric lesions from H+/K+ ATPase-Cre; LKB1^L/L^; PTEN^L/+^, H+/K+ ATPase-Cre; PTEN^L/L^ mice and wild-type controls. Histology of gastric neoplastic lesions from the corpus, antrum, and duodenum and invaded nearby tissues of H^+^/K^+^ ATPase-Cre; LKB1^L/L^; PTEN^L/L^ gastric tumors compared with gastric lesions from H^+^/K^+^ ATPase-Cre; LKB1^L/L^; PTEN^L/+^ and H^+^/K^+^ ATPase-Cre; PTEN^L/L^ mice and normal controls, showing that a loss of LKB1 and PTEN induces gastric adenocarcinoma in mice. Representative hematoxylin and eosin (H&E)-stained sections from the stomachs of normal mice and those with the indicated mutations. Asterisks indicate smooth muscle layers invaded by tumors in the H^+^/K^+^ ATPase-Cre; LKB1^L/L^; PTEN^L/L^ mice. Original magnification (100×) and small insert (400×). Scale bars: 100 μm.

**Figure 4 cancers-15-05893-f004:**
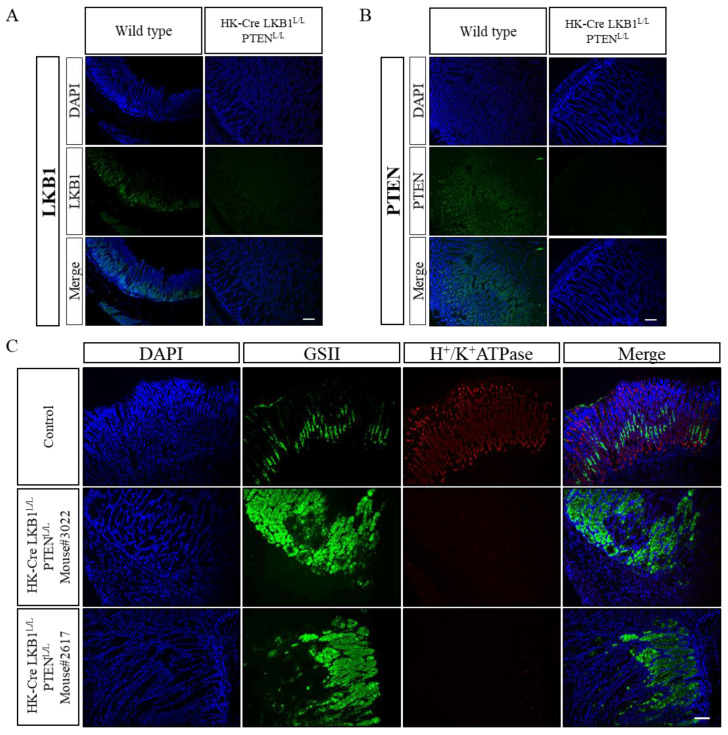
Defects in LKB1 and PTEN disrupt gastric cells distribution and homeostasis. (**A**,**B**) Loss of LKB1 and PTEN protein expression in H+/K+ ATPase-Cre; LKB1^L/L^; PTEN^L/L^ mice, as observed through immunofluorescence analysis. This demonstrates the complete absence of LKB1 and PTEN expression in the gastric epithelium of these mice. (**C**) Immunofluorescence analysis using anti GSII and H+/K+ ATPase antibodies in H+/K+ ATPase-Cre; LKB1^L/L^; PTEN^L/L^ (mouse #3022 and 2617) and control mice. Scale bars: 100 μm.

**Figure 5 cancers-15-05893-f005:**
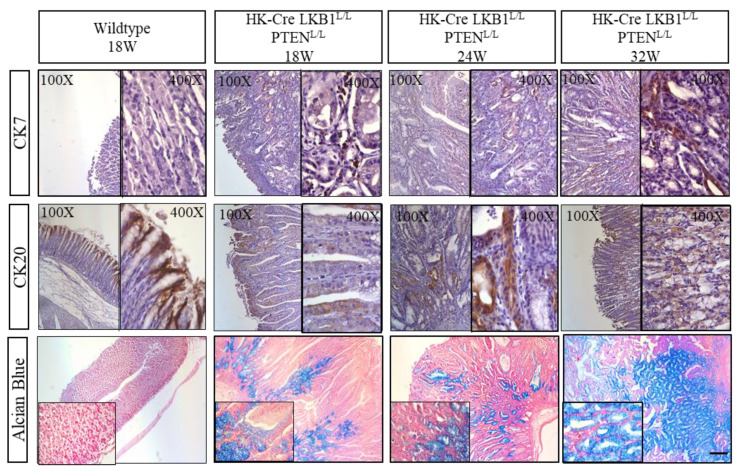
The expression patterns of cytokeratins and Alcian blue staining of gastric tissues derived from H^+^/K^+^ ATPase-Cre; LKB1^L/L^; PTEN^L/L^ and control mice. Comparative staining of gastric neoplastic lesions and normal stomach shows expression of CK7 and CK20 in the majority of the tumors derived from the H+/K+ ATPase-Cre; LKB1^L/L^; PTEN^L/L^ mice (magnification 100× and 400×). Alcian blue staining showing an abundance of acid mucins in the corpus gland and the invasion region of the gastric lesions. Original magnification (100×) and small insert (400×). Scale bars: 100 μm.

**Figure 6 cancers-15-05893-f006:**
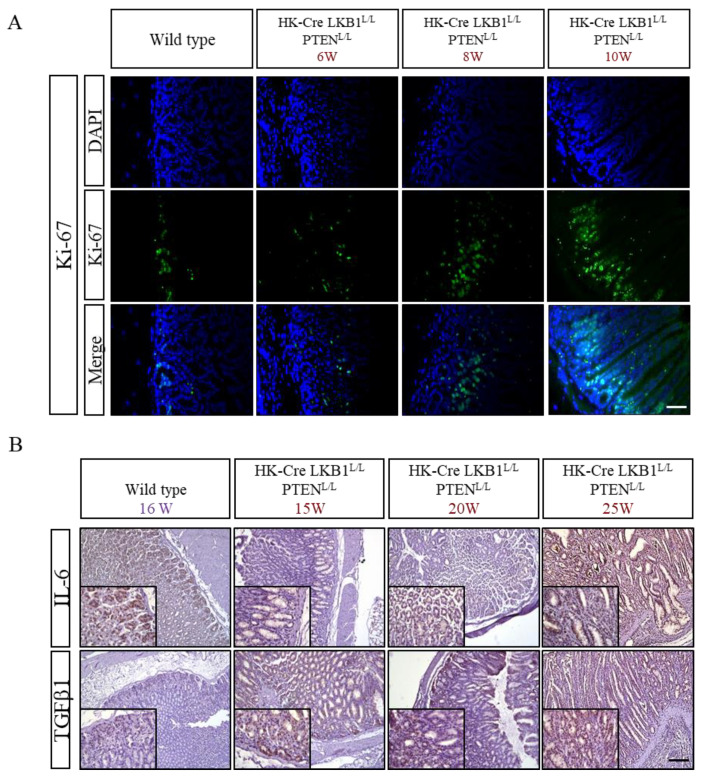
Increased proliferation and inflammation in the gastric lesions of H+/K+ ATPase-Cre; LKB1^L/L^; PTEN^L/L^ mice. (**A**) Immunofluorescence and (**B**) immunohistochemical staining using Ki67, TGF-β1, and IL-6 antibodies in gastric tissues of H+/K+ ATPase-Cre; LKB1^L/L^; PTEN^L/L^ mice compared to normal mouse stomach tissues. Scale bars: 100 μm.

**Figure 7 cancers-15-05893-f007:**
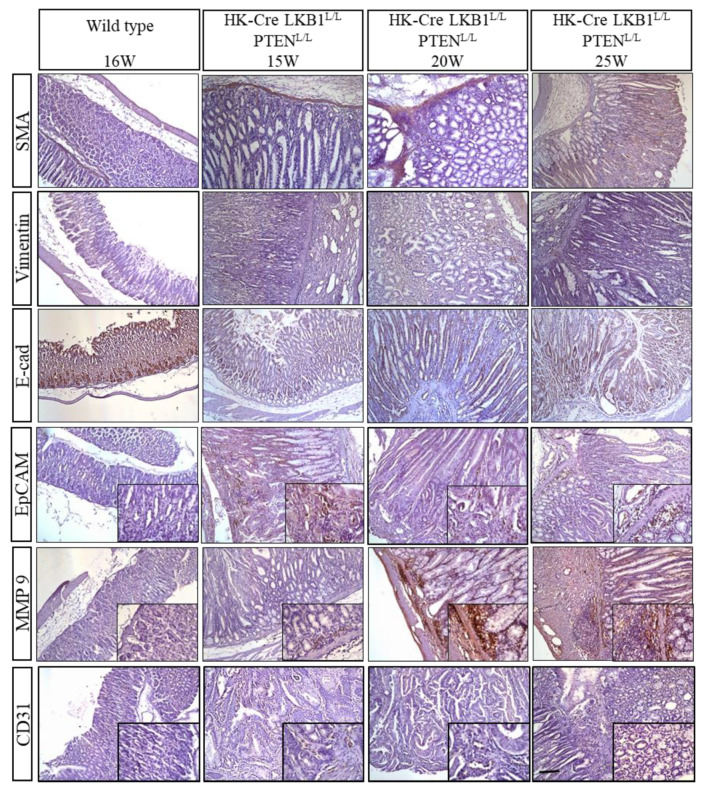
Detection of EMT and CD31 angiogenic marker expression in gastric neoplastic lesions of H+/K+ ATPase-Cre; LKB1^L/L^; PTEN^L/L^ mice. Immunohistochemical analysis revealed that mesenchymal and angiogenic markers were upregulated in gastric lesions of H+/K+ ATPase-Cre; LKB1^L/L^; PTEN^L/L^ mice compared with wild-type controls. Original magnification (100×) and small insert (400×). Scale bars: 100 μm.

**Figure 8 cancers-15-05893-f008:**
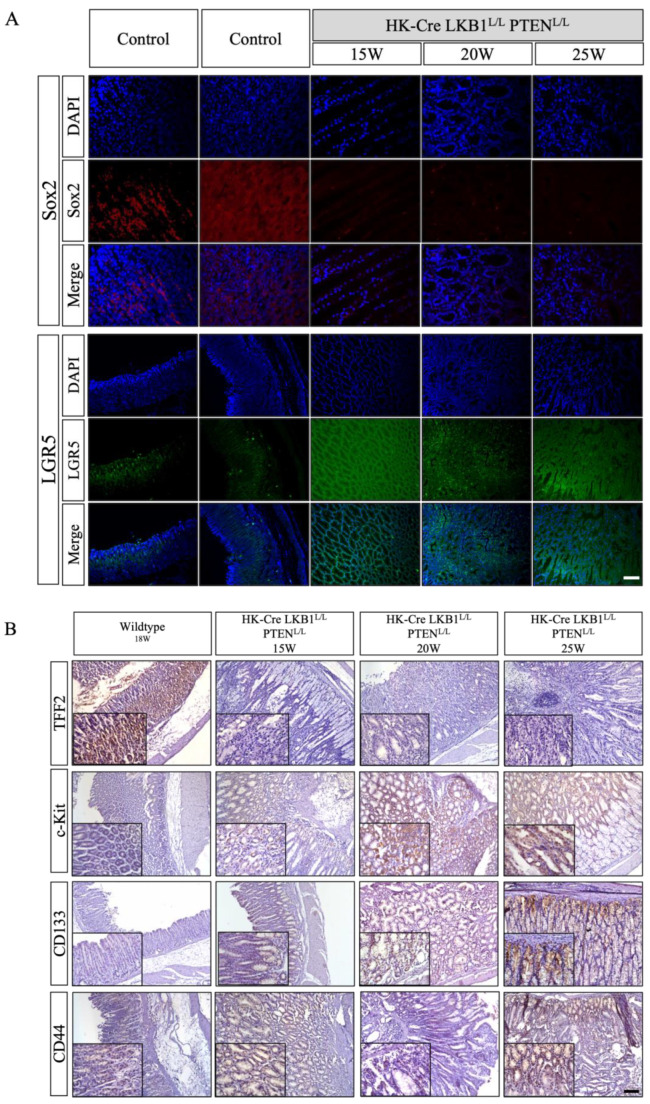
Immunofluorescent and immunohistochemical analyses of SOX2, LGR5, TFF2, CD44, CD133, and c-kit expression in normal stomachs (control; wild-type) and neoplastic gastric lesions of H^+^/K^+^ ATPase-Cre; LKB1^L/L^; PTEN^L/L^ mice. Representative images of immunofluorescent (**A**) and immunohistochemical (**B**) staining in the control and H^+^/K^+^ ATPase-Cre; LKB1^L/L^; PTEN^L/L^ groups. Our results show that the loss of LKB1 and PTEN in the mouse stomach resulted in a reduction in the expression of the differentiation markers SOX2 and TFF2. Conversely, there was an increase in the expression of stem cell markers, including CD44, CD133, c-kit, and LGR5, within the gastric neoplasms of these mice. Original magnification (100×) and small insert (400×). Scale bars: 100 μm.

## Data Availability

The data are available from the corresponding author upon reasonable request.

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
