# Peer review of "Development of a Genetically Engineered Mouse Model Recapitulating LKB1 and PTEN Deficiency in Gastric Cancer Pathogenesis"

_cancers, 2023, doi:10.3390/cancers15245893_

Round 1
Reviewer 1 Report (Previous Reviewer 1)
Comments and Suggestions for Authors
Many parts of this manuscript have been corrected appropriately, but there are still some errors.
1. The 'Pten' in Figure 1 should be written as 'PTEN'. Overall, the marking of gene or protein name must match.
2. The notation in the figure is not constant. For example, it is written as 'wild type' in figure 1, however 'Wild type' in figure 3. You need to check whether the first letter is in capital letters.
3. The content of legend about figure 1D doesn't match the figure. The legend excludes content about 'PTENL/L'.
4. The 20min of line 219 should be modified to 20mins.
5. Please add the picture of the result for ‘PTEN L/L’ in ‘â…µ of Figure 2A’.
6. Only the last picture of 'Antrum' in figure 3 has a red line and the overall picture is not aligned.
7. There is no result figure for H+/K+ 343 ATPase-Cre; LKB1L/L as described in the manuscript and legend of Figure 3.
8. In line 402, the mouse number is marked as #2167 but the figure is marked as #2617. Please match the number.
Comments on the Quality of English LanguageMany parts of this manuscript have been corrected appropriately, but there are still some errors.
1. The 'Pten' in Figure 1 should be written as 'PTEN'. Overall, the marking of gene or protein name must match.
2. The notation in the figure is not constant. For example, it is written as 'wild type' in figure 1, however 'Wild type' in figure 3. You need to check whether the first letter is in capital letters.
3. The content of legend about figure 1D doesn't match the figure. The legend excludes content about 'PTENL/L'.
4. The 20min of line 219 should be modified to 20mins.
5. Please add the picture of the result for ‘PTEN L/L’ in ‘â…µ of Figure 2A’.
6. Only the last picture of 'Antrum' in figure 3 has a red line and the overall picture is not aligned.
7. There is no result figure for H+/K+ 343 ATPase-Cre; LKB1L/L as described in the manuscript and legend of Figure 3.
8. In line 402, the mouse number is marked as #2167 but the figure is marked as #2617. Please match the number.
Author Response
Responses to the Reviewer 1
We thank the editors and reviewers of our manuscript for their helpful comments and suggestions. Please find our point-by-point responses below.
Regarding reviewer 1 comments;
- The 'Pten' in Figure 1 should be written as 'PTEN'. Overall, the marking of gene or protein name must match.
Response: Thank you for bringing this to our attention. We have corrected the naming inconsistency, and 'Pten' in Figure 1 is now appropriately written as 'PTEN'.
- The notation in the figure is not constant. For example, it is written as 'wild type' in figure 1, however 'Wild type' in figure 3. You need to check whether the first letter is in capital letters.
Response: Thank you for pointing out the inconsistency in notation. We have rectified the issue by ensuring that the term "Wild type" now consistently begins with a capital letter in all figures, aligning with your suggestion.
- The content of legend about figure 1D doesn't match the figure. The legend excludes content about 'PTENL/L'.
Response: Thank you so much for bringing this to our attention. We have carefully revised the figure legend for Figure 1D to ensure that it accurately reflects the content of the figure. The discrepancy regarding 'PTENL/L' has been addressed in our revised manuscript.
- The 20min of line 219 should be modified to 20mins.
Response: Thank you for catching that error. We have made the necessary correction, and '20min' in line 219 has been modified to '20mins.' We appreciate your attention to detail.
- Please add the picture of the result for ‘PTEN L/L’ in ‘â…µ of Figure 2A’.
Response: Thank you for the reviewer's comment. In response to the request to add the picture of the result for 'PTEN L/L' in 'â…µ of Figure 2A,' we have incorporated the relevant data into Figure 3. Specifically, we have presented the gross morphology and histology of stomach tissues from H+/K+- ATPase Cre PTENL/L mice.
- Only the last picture of 'Antrum' in figure 3 has a red line and the overall picture is not aligned.
Response: Done. This point is well taken.
- There is no result figure for H+/K+ 343 ATPase-Cre; LKB1L/L as described in the manuscript and legend of Figure 3.
Response: Done. This point is well taken. Thank you for bringing this to our attention. We appreciate your feedback, and we have addressed the issue by revising the legend of Figure 3. The necessary correction has been made to accurately describe H+/K+ ATPase-Cre; PTENL/L, aligning it with the information provided in the manuscript.
- In line 402, the mouse number is marked as #2167 but the figure is marked as #2617. Please match the number.
Response: Done. This point is well taken.
We believe these revisions address the concerns raised during the review process and improve the overall quality and reliability of our manuscript. We appreciate the time and effort invested by the reviewers and the editorial team in evaluating our work. We hope the revised manuscript is acceptable for publication.
With kindest regards
Kuang
12/12/2023

Reviewer 2 Report (Previous Reviewer 2)
Comments and Suggestions for Authors
The authors have improved the manuscript.
Minor points
line 458. Remove the letter e from meanwhiele.
Author Response
Dear Reviewer,
Thank you very much for accepting our manuscript pending suggested minor revision of the article according to the reviewers’ comments. We appreciate the time and effort invested by the reviewers and the editorial team in evaluating our work.
Regarding reviewer's minor point:
line 458. Remove the letter e from meanwhiele.
Response: Done. We have double-checked for this typo and corrected it already.
We hope the revised manuscript is acceptable for publication.
With kindest regards
Kuang
12/12/2023

Reviewer 3 Report (Previous Reviewer 3)
Comments and Suggestions for Authors
I am satisfied that the authors have addressed my points of concern and that this manuscript is now acceptable for publication in Cancers.
Author Response
Dear Reviewer,
We sincerely appreciate the time and effort invested by the reviewers and the editorial team in evaluating our work. Thank you very much for carefully reviewing and accepting our manuscript for publication.
Best regards,
Kuang
12/12/2023

This manuscript is a resubmission of an earlier submission. The following is a list of the peer review reports and author responses from that submission.
Round 1
Reviewer 1 Report
Comments and Suggestions for Authors
"This manuscript demonstrates, through a mouse model, that defects in LKB1 and PTEN promote the proliferation, angiogenesis, and metastasis of gastric cancer. First, the authors constructed a conditional deletion system for LKB1 and PTEN in the mouse stomach. Using this mouse model, the authors confirmed that the deletion of both genes accelerates the growth and metastasis of gastric cancer. Histological data have shown that intestinal-type gastric adenocarcinoma is induced in H+/K+ ATPase-Cre LKB1L/L PTEN L/L mice compared to WT mice. Additionally, in H+/K+ ATPase-Cre LKB1L/L PTEN L/L mice, not only cancer cell proliferation but also metastatic marker expression, angiogenesis, and progression increased. These findings show that LKB1 and PTEN play a crucial role in the overall progression of gastric cancer. However, there are errors and questions in the manuscript that need correction.
- In Figure 1, the descriptions for B and C are reversed. Also, the contents of B and C are reversed in the figure legend. Please ensure the correct labeling of the mouse line used. A comprehensive review of the legend and data in Figure 1 is required.
- The description in lines 238 to 241 should be rearranged in the order of the figures.
- In Figure 1A, 'H/K+ATPase Promoter' should be modified to 'H+/K+ATPase Promoter'.
- Adding a survival time plot, as described in lines 255 to 258, may help in understanding Figure 2.
- There is no comparison data for the H+/K+ATPase-Cre;LKB1L/L mice referred to in line 257. You must add data or label it as 'data not shown'.
- The term used in line 257 as 'H+/K+ATPase mice' is marked as 'wild type' in the figure data. You need to reorganize and modify this terminology.
- In line 268, add 'Figure 3Aiii' at the end of the sentence.
- The authors mention seeing 'eyes and tails' on line 270, but there is only a tail photo in Figure 2A-vi. Confirmation of the data is required.
- The statement in line 274 refers to Figure 2B, not Figure 2A-iv. Please confirm and make the necessary modifications. Additionally, ensure accurate reference to Figure 2A-iv in line 276.
- Adjust the line spacing in the CBC part of the table in Figure 2B.
- In Figure 3, the data and descriptions need to be accurately matched. Modify the description and legend for Figure 3.
- Add a scale bar to Figure 4A and B. Explain what 3022 and 2617 mean in Figure 4B.
- Arrange the pictures in Figure 5 consistently, and ensure uniform border thickness.
- In the legend of Figure 6, change A and B to (A) and (B) for consistency with other legend cases.
- Provide additional explanations for the red markings in Figure 7.
- For the description in lines 442-444, present the correct data numbers. For example, describe the SOX2 data in Figure 8A and the TFF2 data in Figure 8B separately.
- Ensure that the picture spacing in Figure 8A is correct, and add a scale bar to Figure 8B.
- Verify the consistency of abbreviations and notations. For example, 'gastric cancer' should be modified to 'G.C.' in lines 54 and 60. Check 'min' in lines 151 and 208. Correct 'allcian blue' to 'alcian blue' in Figure 5. Remove hyphens from words like 'significant' in line 272 and 'assessments' in line 276. Add the full name for CSC in line 449."
Comments on the Quality of English Language
"This manuscript demonstrates, through a mouse model, that defects in LKB1 and PTEN promote the proliferation, angiogenesis, and metastasis of gastric cancer. First, the authors constructed a conditional deletion system for LKB1 and PTEN in the mouse stomach. Using this mouse model, the authors confirmed that the deletion of both genes accelerates the growth and metastasis of gastric cancer. Histological data have shown that intestinal-type gastric adenocarcinoma is induced in H+/K+ ATPase-Cre LKB1L/L PTEN L/L mice compared to WT mice. Additionally, in H+/K+ ATPase-Cre LKB1L/L PTEN L/L mice, not only cancer cell proliferation but also metastatic marker expression, angiogenesis, and progression increased. These findings show that LKB1 and PTEN play a crucial role in the overall progression of gastric cancer. However, there are errors and questions in the manuscript that need correction.
- In Figure 1, the descriptions for B and C are reversed. Also, the contents of B and C are reversed in the figure legend. Please ensure the correct labeling of the mouse line used. A comprehensive review of the legend and data in Figure 1 is required.
- The description in lines 238 to 241 should be rearranged in the order of the figures.
- In Figure 1A, 'H/K+ATPase Promoter' should be modified to 'H+/K+ATPase Promoter'.
- Adding a survival time plot, as described in lines 255 to 258, may help in understanding Figure 2.
- There is no comparison data for the H+/K+ATPase-Cre;LKB1L/L mice referred to in line 257. You must add data or label it as 'data not shown'.
- The term used in line 257 as 'H+/K+ATPase mice' is marked as 'wild type' in the figure data. You need to reorganize and modify this terminology.
- In line 268, add 'Figure 3Aiii' at the end of the sentence.
- The authors mention seeing 'eyes and tails' on line 270, but there is only a tail photo in Figure 2A-vi. Confirmation of the data is required.
- The statement in line 274 refers to Figure 2B, not Figure 2A-iv. Please confirm and make the necessary modifications. Additionally, ensure accurate reference to Figure 2A-iv in line 276.
- Adjust the line spacing in the CBC part of the table in Figure 2B.
- In Figure 3, the data and descriptions need to be accurately matched. Modify the description and legend for Figure 3.
- Add a scale bar to Figure 4A and B. Explain what 3022 and 2617 mean in Figure 4B.
- Arrange the pictures in Figure 5 consistently, and ensure uniform border thickness.
- In the legend of Figure 6, change A and B to (A) and (B) for consistency with other legend cases.
- Provide additional explanations for the red markings in Figure 7.
- For the description in lines 442-444, present the correct data numbers. For example, describe the SOX2 data in Figure 8A and the TFF2 data in Figure 8B separately.
- Ensure that the picture spacing in Figure 8A is correct, and add a scale bar to Figure 8B.
- Verify the consistency of abbreviations and notations. For example, 'gastric cancer' should be modified to 'G.C.' in lines 54 and 60. Check 'min' in lines 151 and 208. Correct 'allcian blue' to 'alcian blue' in Figure 5. Remove hyphens from words like 'significant' in line 272 and 'assessments' in line 276. Add the full name for CSC in line 449."
Author Response
Date: 11/25/2023
Rebuttal letter
Dear Editors,
Thank you very much for providing a comprehensive review of our manuscript. We are pleased to resubmit an extensively revised manuscript with some corrections here. On behalf of my co-authors, we would like to express our great appreciation to editor and reviewers. We greatly appreciate the efficient and professional processing of our paper by your team. As for the reviewers’ comments, we have tried our best to revise our manuscript according to their comments. We have studied reviewer’s comments carefully and have made revision which marked in red in the manuscript. Below, we have addressed all of the points raised by the reviewers.
Review 1
Comments and Suggestions for Authors
"This manuscript demonstrates, through a mouse model, that defects in LKB1 and PTEN promote the proliferation, angiogenesis, and metastasis of gastric cancer. First, the authors constructed a conditional deletion system for LKB1 and PTEN in the mouse stomach. Using this mouse model, the authors confirmed that the deletion of both genes accelerates the growth and metastasis of gastric cancer. Histological data have shown that intestinal-type gastric adenocarcinoma is induced in H+/K+ ATPase-Cre LKB1L/L PTEN L/L mice compared to WT mice. Additionally, in H+/K+ ATPase-Cre LKB1L/L PTEN L/L mice, not only cancer cell proliferation but also metastatic marker expression, angiogenesis, and progression increased. These findings show that LKB1 and PTEN play a crucial role in the overall progression of gastric cancer. However, there are errors and questions in the manuscript that need correction.
- In Figure 1, the descriptions for B and C are reversed. Also, the contents of B and C are reversed in the figure legend. Please ensure the correct labeling of the mouse line used. A comprehensive review of the legend and data in Figure 1 is required.
Response: Thank you very much for your carefully reading of the manuscript. The points are well taken.
- The description in lines 238 to 241 should be rearranged in the order of the figures.
Response: The reviewer’s point is well taken. We have amended this text accordingly.
- In Figure 1A, 'H/K+ATPase Promoter' should be modified to 'H+/K+ATPase Promoter'.
Response: Done. The reviewer’s point is well taken.
- Adding a survival time plot, as described in lines 255 to 258, may help in understanding Figure 2.
Response: The reviewer’s point is well taken. We added a survival time plot Figure 1D in the end of the sentence to help in understanding Figure 2.
- There is no comparison data for the H+/K+ATPase-Cre;LKB1L/L mice referred to in line 257. You must add data or label it as 'data not shown'.
Response: Done. This point is well taken
.
- The term used in line 257 as 'H+/K+ATPase mice' is marked as 'wild type' in the figure data. You need to reorganize and modify this terminology.
Response: Done. This point is well taken.
- In line 268, add 'Figure 3Aiii' at the end of the sentence.
Response: Done. The reviewer’s point is well taken.
- The authors mention seeing 'eyes and tails' on line 270, but there is only a tail photo in Figure 2A-vi. Confirmation of the data is required.
Response: The reviewer’s point is well taken. We removed “eyes” description.
- The statement in line 274 refers to Figure 2B, not Figure 2A-iv. Please confirm and make the necessary modifications. Additionally, ensure accurate reference to Figure 2A-iv in line 276.
Response: The reviewer’s point is well taken. We have amended this text accordingly, and cited an accurate reference on it.
- Adjust the line spacing in the CBC part of the table in Figure 2B.
Response: Done. The reviewer’s point is well taken.
- In Figure 3, the data and descriptions need to be accurately matched. Modify the description and legend for Figure 3.
Response: The reviewer’s point is well taken. We have amended this text accordingly. We modified the description and legend for Figure 3.
- Add a scale bar to Figure 4A and B. Explain what 3022 and 2617 mean in Figure 4B.
Response: Done. The reviewer’s point is well taken.
- Arrange the pictures in Figure 5 consistently, and ensure uniform border thickness.
Response: The reviewer’s point is well taken. We have amended this text accordingly.
- In the legend of Figure 6, change A and B to (A) and (B) for consistency with other legend cases.
Response: The reviewer’s point is well taken.
- Provide additional explanations for the red markings in Figure 7.
Response: The reviewer’s point is well taken. We removed the red markings in the Figure 7.
- For the description in lines 442-444, present the correct data numbers. For example, describe the SOX2 data in Figure 8A and the TFF2 data in Figure 8B separately.
Response: This point is well taken.
- Ensure that the picture spacing in Figure 8A is correct, and add a scale bar to Figure 8B.
Response: The reviewer's feedback has been considered, and appropriate modifications have been made.
- Verify the consistency of abbreviations and notations. For example, 'gastric cancer' should be modified to 'G.C.' in lines 54 and 60. Check 'min' in lines 151 and 208. Correct 'allcian blue' to 'alcian blue' in Figure 5. Remove hyphens from words like 'significant' in line 272 and 'assessments' in line 276. Add the full name for CSC in line 449."
Response: All reviewer's inputs here have been taken into account, and necessary adjustments (corrections) have been implemented.
We greatly appreciate the reviewers for their helpful comments. We hope the revised manuscript is acceptable for publication.
With kindest regards
Yours Sincerely
Kuang-Hung Cheng
Kuang-hung Cheng, Ph.D.
Chair and Professor of Biomedical Science Institute
National Sun Yat-Sen University
Kaohsiung, Taiwan 807
TEL: 886-7-5252000 ext 5817
Fax: 886-7-5250197
Email: khcheng@faculty.nsysu.edu.tw

Reviewer 2 Report
Comments and Suggestions for Authors
The authors describe a mouse model wherein LKB1 alleles are rendered inactive, alongside conditional PTEN null alleles, driven by the stomach-specific H+/K+ ATPase-Cre transgenic line.
Comments
1- The introduction is long-winded, particularly regarding the description of the mouse stomach. Likewise, the discussion should focus more on PTEN and STK11 in gastric cancer and avoid digressions.
2- Lines 83-107. STK11 is described for its function and what is known in tumors in general. What do we know about STK11 and gastric cancer?
3- Can you provide the changes in STK11 and PTEN expression in mice development?
4- Have you verified the involvement of other organs besides the stomach of the transgenic mice?
5- You should explain what is known about the importance of STK11 and PTEN in gastric cancer. What is the combined genetic alteration rate of STK11 and PTEN in gastric cancer?
6- lines 495-500. Are you saying here that STK11/PTEN null transgenic mice are actually a model for gastric cancer involving STK11, PTEN, and ATP4B?
7- How did you explain the set of immune effects and emerging changes in transgenic compared to wt mice? Can you rule out that genetic manipulation has altered hematopoiesis?
8- References should be updated to the most recent articles. The oldest reference is from 2019.
Comments on the Quality of English LanguageThe manuscript reads well.
Author Response
Date 11/23/2023
Rebuttal letter
Dear Editors,
Thank you very much for providing a comprehensive review of our manuscript. We are pleased to resubmit an extensively revised manuscript with some corrections here. On behalf of my co-authors, we would like to express our great appreciation to editor and reviewers. We greatly appreciate the efficient and professional processing of our paper by your team. As for the reviewers’ comments, we have tried our best to revise our manuscript according to their comments. We have studied reviewer’s comments carefully and have made revision which marked in red in the manuscript. Below, we have addressed all of the points raised by the reviewers.
Review 2
Comments and Suggestions for Authors
The authors describe a mouse model wherein LKB1 alleles are rendered inactive, alongside conditional PTEN null alleles, driven by the stomach-specific H+/K+ ATPase-Cre transgenic line.
Response: We thank the reviewer for his/her comments and suggestions. We have incorporated nearly every suggestion in the revised version as summarized below.
Comments
1- The introduction is long-winded, particularly regarding the description of the mouse stomach. Likewise, the discussion should focus more on PTEN and STK11 in gastric cancer and avoid digressions.
Response: Thank you for the constructive suggestions from the reviewer. In response, our revised manuscript streamlines the introduction on the anatomical features of the mouse stomach. Moreover, we have enriched the content by providing more comprehensive descriptions of the roles played by PTEN and STK11 in gastric cancer.
2- Lines 83-107. STK11 is described for its function and what is known in tumors in general. What do we know about STK11 and gastric cancer?
Response: We appreciate the reviewers' valuable feedback, enhancing the quality of our manuscript. In response, we have enhanced the content by offering more comprehensive descriptions of the roles played by STK11 in gastric cancer, particularly in the discussion section of our revised manuscript.
3- Can you provide the changes in STK11 and PTEN expression in mice development?
Response: Thanks for the constructive suggestion from the reviewer. In response, we have included descriptions of the developmental defects observed in LKB1/STK11 and PTEN knockout mice in the discussion section of our revised manuscript.
4- Have you verified the involvement of other organs besides the stomach of the transgenic mice?
Response: Yes, we conducted Cre and H+/K+ ATPase IHC staining in diverse mouse tissues, validating exclusive Cre expression in stomach parietal cells, consistent with previously published results.
Reference: Zengming Zhao, Ning Hou, Yanxun Sun, Yan Teng, Xiao Yang. Atp4b promoter directs the expression of Cre recombinase in gastric parietal cells of transgenic mice. J Genet Genomics. 2010 Sep;37(9):647-52.
5- You should explain what is known about the importance of STK11 and PTEN in gastric cancer. What is the combined genetic alteration rate of STK11 and PTEN in gastric cancer?
Response: This point is well taken. We enriched the content by offering more comprehensive descriptions of the roles played by PTEN and STK11 in gastric cancer in our revised manuscript.
6- lines 495-500. Are you saying here that STK11/PTEN null transgenic mice are actually a model for gastric cancer involving STK11, PTEN, and ATP4B?
Response: "In lines 495-500, we discussed cases of patients with defects in the H+/K+ ATPase alpha subunit gene in the stomach, leading to impaired vitamin B absorption and resulting in iron-deficiency anemia. This finding is consistent with our H+/K+ ATPase gastric cancer mouse model, where engineered gene alterations in the parietal cells of the stomach not only disrupt vitamin B absorption but also drive gastric tumorigenesis, ultimately contributing to the development of anemia."
7- How did you explain the set of immune effects and emerging changes in transgenic compared to wt mice? Can you rule out that genetic manipulation has altered hematopoiesis?
Response: "Thank you for highlighting this important point in your review. It's crucial to note that the Cre/loxp system utilized in our study is under the control of the H+/K+ ATPase pump gene (Atp4b), which specifically expresses in the parietal cells of the stomach. Consequently, any gene editing or modification driven by the H+/K+ ATPase–Cre will exclusively occur in the parietal cells of the stomach."
Reference: Zengming Zhao, Ning Hou, Yanxun Sun, Yan Teng, Xiao Yang. Atp4b promoter directs the expression of Cre recombinase in gastric parietal cells of transgenic mice. J Genet Genomics. 2010 Sep;37(9):647-52.
8- References should be updated to the most recent articles. The oldest reference is from 2019.
Response: Thank you for your feedback. In our revised manuscript, we have incorporated citations from more recent articles to enhance the current literature review.
We greatly appreciate the reviewers for their helpful comments. We hope the revised manuscript is acceptable for publication.
With kindest regards
Yours Sincerely
Kuang-Hung Cheng
Kuang-hung Cheng, Ph.D
Chair and Professor of Biomedical Science Institute
National Sun Yat-Sen University
Kaohsiung, Taiwan 807
TEL: 886-7-5252000 ext 5817
Fax: 886-7-5250197
Email: khcheng@faculty.nsysu.edu.tw

Reviewer 3 Report
Comments and Suggestions for Authors
The authors have created a new mouse model for gastric cancer using conditional knockout of LKB1 and PTEN. By 20 weeks post induction the double knock out mice display marked gastric hyperplasia and gastric cancer. In this study, the authors have characterised the histology of the stomachs in their model as well as examining the expression and localisation of a number of key factors in the development of gastric carcinogenesis. The use of the Atp4b promoter has been demonstrated in other models of gastric carcinoma, but the gastric specific double knock out LKB1 and PTEN is novel and of interest. An additional benefit in this model is that there is quicker adenocarcinoma formation than another published (Shimada et al, 2012). However, I couldn't easily glean what the penetrance of this model is??
In general, this is an interesting study which presents a novel model of gastric specific compound mutation of LKBR and PTEN. After addressing the following points I think this manuscript would be appropriate for publication in Cancers.
Minor points:
- Line 239: Says limited "survival period of 20 weeks" but this is contradicted by next sentence "notably at 20 weeks the mice start to develop hyperplasia" (line 255).
- Line 403: Meanwhile has a typo
Major points:
- In all figures containing images of stomachs there should be an age-matched WT image. For example: Figure 5, there is WT 18 weeks, then a progression of Double knock out mice up to 32 weeks old. In my opinion there should be age appropriate WT images for comparison. Additional images could be placed in supplementary files.
- I feel that there should be clearer presentation or description of the histology figures. For non histologists, the figures are quite overwhelming and for example it would be good to actually highlight where the parietal cells are on images or explain what to look for in the images.
- In figure 5, what is the purple counterstain in the CK7 and CK20 images? Why has a nuclear counterstain (such as H & E) not been utilised?
Comments on the Quality of English LanguageIn general, the English is good. There are a couple of minor typos to be corrected, which another round of proofreading should identify.
Author Response
Rebuttal letter
Dear Editors,
Thank you very much for providing a comprehensive review of our manuscript. We are pleased to resubmit an extensively revised manuscript with some corrections here. On behalf of my co-authors, we would like to express our great appreciation to editor and reviewers. We greatly appreciate the efficient and professional processing of our paper by your team. As for the reviewers’ comments, we have tried our best to revise our manuscript according to their comments. We have studied reviewer’s comments carefully and have made revision which marked in red in the manuscript. Below, we have addressed all of the points raised by the reviewers.
Review 3
Comments and Suggestions for Authors
The authors have created a new mouse model for gastric cancer using conditional knockout of LKB1 and PTEN. By 20 weeks post induction the double knock out mice display marked gastric hyperplasia and gastric cancer. In this study, the authors have characterised the histology of the stomachs in their model as well as examining the expression and localisation of a number of key factors in the development of gastric carcinogenesis. The use of the Atp4b promoter has been demonstrated in other models of gastric carcinoma, but the gastric specific double knock out LKB1 and PTEN is novel and of interest. An additional benefit in this model is that there is quicker adenocarcinoma formation than another published (Shimada et al, 2012). However, I couldn't easily glean what the penetrance of this model is??
In general, this is an interesting study which presents a novel model of gastric specific compound mutation of LKBR and PTEN. After addressing the following points I think this manuscript would be appropriate for publication in Cancers.
Minor points:
- Line 239: Says limited "survival period of 20 weeks" but this is contradicted by next sentence "notably at 20 weeks the mice start to develop hyperplasia" (line 255).
Response: "Thank you for the careful revision of our manuscript. We have addressed the mentioned typo and adjusted the timeframe to 30 weeks in accordance with their survival curve."
Line 403: Meanwhile has a typo
Response: Thank you for bringing that to our attention. Done. We correct the typo in the revised manuscript.
Major points:
- In all figures containing images of stomachs there should be an age-matched WT image. For example: Figure 5, there is WT 18 weeks, then a progression of Double knock out mice up to 32 weeks old. In my opinion there should be age appropriate WT images for comparison. Additional images could be placed in supplementary files.
Response: We sincerely appreciate the reviewer's valuable suggestions. However, we would like to clarify that we included 18-week normal stomach samples as controls for the 18-week HK CreLKB1L/L PTENL/L mice. Subsequently, we compared the progression of malignancy in HK CreLKB1L/L PTENL/L mice at 25 and 32 weeks to the 18-week group.
- I feel that there should be clearer presentation or description of the histology figures. For non histologists, the figures are quite overwhelming and for example it would be good to actually highlight where the parietal cells are on images or explain what to look for in the images.
Response: We incorporated small inserts for high-power views in each IHC dataset. Distinguishing parietal cells becomes challenging after the onset of gastric tumorigenesis, as they actively participate in and may undergo transformation into tumor cells.
- In figure 5, what is the purple counterstain in the CK7 and CK20 images? Why has a nuclear counterstain (such as H & E) not been utilised?
Response: Done. Hematoxylin is a dye that stains cell nuclei blue or purple. We utilized Hematoxylin as a counterstain to visualize cell nuclei. Subsequently, we demonstrated the presence of CK7 and CK20 staining in the cytosol after examining the IHC figures under high power. we included high-power views of CK7 and CK20 IHC images in our revised manuscript.
We greatly appreciate the reviewers for their helpful comments. We hope the revised manuscript is acceptable for publication.
With kindest regards
Yours Sincerely
Kuang-Hung Cheng
Kuang-hung Cheng, Ph.D
Chair and Professor of Biomedical Science Institute
National Sun Yat-Sen University
Kaohsiung, Taiwan 807
TEL: 886-7-5252000 ext 5817
Fax: 886-7-5250197
Email: khcheng@faculty.nsysu.edu.tw
